# Role of mTOR through Autophagy in Esophageal Cancer Stemness

**DOI:** 10.3390/cancers14071806

**Published:** 2022-04-01

**Authors:** Liang Du, Da Wang, Peter W. Nagle, Andries A. H. Groen, Hao Zhang, Christina T. Muijs, John Th. M. Plukker, Robert P. Coppes

**Affiliations:** 1Section Molecular Cell Biology, Department of Biomedical Sciences of Cells and Systems, University Medical Center Groningen, University of Groningen, 9700 RB Groningen, The Netherlands; l.du@umcg.nl (L.D.); d.wang@erasmusmc.nl (D.W.); peter.nagle@ed.ac.uk (P.W.N.); a.h.groen-4@umcutrecht.nl (A.A.H.G.); 2Department of Radiation Oncology, University Medical Center Groningen, University of Groningen, 9700 RB Groningen, The Netherlands; c.t.muijs@umcg.nl; 3Graduate School, Shantou University Medical College, Shantou 515041, China; 4Department of Surgery, University Medical Center Groningen, University of Groningen, 9700 RB Groningen, The Netherlands; j.t.m.plukker@umcg.nl; 5Medical Research Council (MRC) Centre for Reproductive Health, The Queen’s Medical Research Institute, The University of Edinburgh, Edinburgh EH16 4TJ, UK; 6Department of Pathology, Institute of Precision Cancer Medicine and Pathology, School of Medicine, Jinan University, Guangzhou 510632, China; haozhang@jnu.edu.cn; 7Department of General Surgery, First Affiliated Hospital of Jinan University, Guangzhou 510632, China

**Keywords:** esophageal cancer, cancer stem cells, mTOR, autophagy, hypoxia

## Abstract

**Simple Summary:**

Esophageal cancer (EC) is a highly aggressive disease with a poor prognosis, which seems related to esophageal cancer stem-like cells (CSCs), which reside in a hypoxic niche. We demonstrated, using EC cell lines and patient-derived organoids, that the hypoxia-responding mammalian target of rapamycin (mTOR) can suppress autophagy and stemness of esophageal CSCs. In addition, mTOR inhibitor Torin-1-mediated CSCs upregulation was significantly reduced in cells treated with autophagy inhibitor, hydroxychloroquine (HCQ). Collectively, our data suggest that autophagy may play a crucial role in mTOR-mediated CSCs repression. The mTOR pathway could be a novel therapeutic target for putative esophageal CSCs.

**Abstract:**

Esophageal cancer (EC) is a highly aggressive disease with a poor prognosis. Therapy resistance and early recurrences are major obstacles in reaching a better outcome. Esophageal cancer stem-like cells (CSCs) seem tightly related with chemoradiation resistance, initiating new tumors and metastases. Several oncogenic pathways seem to be involved in the regulation of esophageal CSCs and might harbor novel therapeutic targets to eliminate CSCs. Previously, we identified a subpopulation of EC cells that express high levels of CD44 and low levels of CD24 (CD44^+^/CD24^−^), show CSC characteristics and reside in hypoxic niches. Here, we aim to clarify the role of the hypoxia-responding mammalian target of the rapamycin (mTOR) pathway in esophageal CSCs. We showed that under a low-oxygen culture condition and nutrient deprivation, the CD44^+^/CD24^−^ population is enriched. Since both low oxygen and nutrient deprivation may inhibit the mTOR pathway, we next chemically inhibited the mTOR pathway using Torin-1. Torin-1 upregulated SOX2 resulted in an enrichment of the CD44^+^/CD24^−^ population and increased sphere formation potential. In contrast, stimulation of the mTOR pathway using MHY1485 induced the opposite effects. In addition, Torin-1 increased autophagic activity, while MHY1485 suppressed autophagy. Torin-1-mediated CSCs upregulation was significantly reduced in cells treated with autophagy inhibitor, hydroxychloroquine (HCQ). Finally, a clearly defined CD44^+^/CD24^−^ CSC population was detected in EC patients-derived organoids (ec-PDOs) and here, MHY1485 also reduced this population. These data suggest that autophagy may play a crucial role in mTOR-mediated CSCs repression. Stimulation of the mTOR pathway might aid in the elimination of putative esophageal CSCs.

## 1. Introduction

Esophageal cancer (EC) is an aggressive disease, with an estimated 604,100 new cases and 544,076 deaths annually, accounting for 3.1% of cancer cases and 5.5% of cancer deaths worldwide [1]. Due to a relatively asymptomatic rapid growth, EC is often presented in an advanced stage with metastatic spread, either lymphatic and/or hematogenous [2]. Unfortunately, only <50% of the newly diagnosed patients are eligible for therapy, with curative intent consisting of neoadjuvant chemoradiation (nCRT), followed by surgery [3]. This trimodality approach has improved survival significantly, with a 5-year survival of 47%, compared to 33% with surgery only [4]. However, only 29% of patients who receive nCRT will reach a pathological complete response, with no vital tumor at the pathological examination of the resection specimen after nCRT [4]. This means that around 70% of all patients receiving nCRT will not respond adequately, of which 18% do not respond at all to the given treatment [4]. Early recurrences occur frequently, even in patients who had complete pathological response after nCRT [4]. This suggests that there are therapy-resistant tumor residues at the primary site or remaining small metastatic disease not detected by conventional imaging or standard pathological examination. Apparently, therapy-resistant tumor cells are a major problem in the treatment of EC. Enhancing tumor sensitivity by targeting resistant cells may reveal new therapeutic options in EC.

Previously, we identified radiation-resistant cells in OE21 esophageal squamous cell carcinoma (ESCC) and OE33 esophageal adenocarcinoma (EAC) cell lines, and in pretreatment biopsies [5]. These resistant cells are marked by a high expression of the cell surface marker CD44 but lack expression of another cell surface marker CD24 (CD44^+^/CD24^−^). These CD44^+^/CD24^−^ cells show cancer stem-like cell (CSC) characteristics, such as radiation resistance, enhanced self-renewal potential, differentiation capacity, and a superior generation of tumors in animal hosts [5]. Interestingly, xenograft tumors show co-localization of CD44^+^/CD24^−^ expression with pimonidazole, a marker for hypoxia, indicating that this population thrives in an oxygen-deprived environment and, thus, potentially contribute to an aggressive phenotype [5]. Indeed, tumor cells that reside under hypoxic conditions have been associated with reduced chemoradiation sensitivity, enhanced tumor growth and the development of metastasis [6]. Several oncogenic pathways respond to hypoxia, including the mammalian target of the rapamycin (mTOR) pathway [7,8,9,10]. The mTOR pathway, as a serine/threonine kinase is a master regulator of cellular processes, leading to protein synthesis, cell growth, proliferation, differentiation, and survival. It maintains cellular homeostasis and is inhibited upon nutrient, growth factor and oxygen deprivation [9,10]. In cancer, the mTOR pathway is a key regulator and controller of the same cellular processes mentioned above, leading to treatment resistance [11]. It may contribute either positively or negatively to a CSC phenotype, depending on the tumor type and the tumor microenvironment [12]. Autophagy is one of the downstream processes related to CSCs [13,14], and is negatively regulated by mTOR [15]. Autophagy is an evolutionary conserved process of eukaryotic cells, designed to serve as a survival mechanism. Unwanted cell components are captured by intracellular double membraned structures or autophagosomes and degraded by lysosomal hydrolase after fusion with lysosomes [16], during cellular stresses, including starvation, hypoxia, pathogen invasion and chemoradiation to maintain cellular homeostasis and provide energy [17,18]. In the context of autophagy regulation, mTOR, as a negative regulator of autophagy, was suppressed by the 5′adenosinemonophosphate-activated protein kinase (AMPK) pathway, which emerges as an activator of autophagy [19]. Therefore, the mTOR pathway and its downstream cellular process of autophagy might be implicated in the regulation of cancer stemness in EC and may open doors to novel targeted therapies. Therefore, we aim to investigate the, thus far, unexplored role of the mTOR pathway in controlling the CD44^+^/CD24^−^ CSC pool in EC.

## 2. Results

### 2.1. Oxygen and Nutrition Influence the CSC Pool

To further validate CD44^+^/CD24^−^ as a putative CSC population in EC, we studied the protein expression of the pluripotent stem cell marker SOX2 in OE33 and OE21 cells, which derived from EAC and ESCC, respectively. Indeed, the EC CD44^+^/CD24^−^ population had a higher SOX2 expression compared to the CD44^+^/CD24^+^ population (Appendix A and Figure 1A). Since our previous results indicate that the EC CD44^+^/CD24^−^ population co-localizes with the area of hypoxia [5], we exposed esophageal adenocarcinoma cell line OE33, esophageal adenocarcinoma cell line BE3 and esophageal squamous cell carcinoma cell line OE21 to reduced oxygen levels of 5% O_2_ for 48 h, compared to the normally used 21%. Although oxygen levels of 5% oxygen are higher than in hypoxia, the percentage of CD44^+^/CD24^−^ was still significantly enriched in reduced oxygen conditions when compared to “normoxic” conditions (21%) (Appendix A and Figure 1B–D), in all tested cell lines, indicating that reducing oxygen enriches CD44^+^/CD24^−^ cells. In tumors, hypoxia may cause poor vascularization, which often also results in a lack of nutrition [20]. To assess whether the nutrition status may affect the putative CD44^+^/CD24^−^ CSC pool, EC cells were starved for 48 h with a starvation medium (HBSS solution containing 1% rich medium) in normoxic conditions. The reduced oxygen conditions resulted in an even more pronounced effect on the percentage of CD44^+^/CD24^−^ cells (Figure 1E,F), showing that reduced oxygen and nutrient deprivation may induce enrichment in putative CSCs. Since the mTOR pathway may respond to changes in both oxygen level and nutritional status [9], we reasoned that this pathway might be involved in the observed selection for the CD44^+^/CD24^−^ cell population. As mTOR is activated through phosphorylation (*p*-mTOR), we measured the phosphorylation status of mTOR and, indeed, *p*-mTOR was lowered, whereas SOX2 was increased under reduced nutrients (Figure 1G). These results suggest that the mTOR pathway might play a role in the maintenance of CSCs.

### 2.2. The mTOR Pathway Negatively Regulates Cancer Stemness and Autophagy

To explore the role of the mTOR pathway in regulating EC stemness, we first examined the effect of adding different concentrations of mTOR inhibitor Torin-1 on cell viability (Appendix A). The concentrations of Torin-1, which reduced less than 50% of the cell viability, were used in further experiments. OE33, BE3 and OE21 cells were treated for 48 h with Torin-1 at 4 nM and 8 nM, doses which have the relatively lowest toxicity for that specific cell line, respectively. All exhibited an expected decrease in *p*-mTOR expression without affecting the total mTOR content and upregulated SOX2 protein expression (Figure 2A–C). Moreover, Torin-1 treatment induced an increase in the percentage of CD44^+^/CD24^−^ in the whole population (Figure 2D–F). The ability to generate spheres in 3D culture is a hallmark of cancer stemness, which was also shown before for CD44^+^/CD24^−^ EC cells [5]. Indeed, when cells were cultured in 3D to generate spheres, Torin-1 treatment resulted in an increased sphere formation when compared to DMSO controls (Figure 2G–I), indicating that inhibition of the mTOR pathway may increase EC cell stemness.

To investigate the role of autophagy, a cellular process associated with cancer stemness [13,21], in the mTOR-related increase in EC cell stemness, we assessed the expression of *p*-mTOR and LC-3II/LC-3I ratio, as an index for autophagosome formation [22,23,24,25]. LC-3I and LC-3II are light chain proteins associated with the formation of autophagolysosomes. Torin-1-treated EC cells exhibited an accumulated LC-3II/LC-3I ratio, suggesting that autophagy is indeed activated (Figure 2J–L).

To verify the involvement of the mTOR pathway in EC stemness, we stimulated mTOR using MHY1485 (a synthesized inhibitor of lysosomal fusion), targeting the mTOR complex 1 (mTORC1). The concentration of MHY1485, which inhibited less than 50% cell viability (Appendix A), was used for subsequent experiments. As expected, OE33, BE3 and OE21 cells displayed a decrease in SOX2 levels and increased *p*-mTOR levels, without affecting total mTOR content (Figure 3A–C), and a decrease in CD44^+^/CD24^−^ CSC population (Figure 3D–F) with MHY1485 compared to DMSO controls. In addition, MHY1485-treated EC cells showed a significant reduction in the number of spheres formed in 3D culture (Figure 3G–I). Furthermore, MHY1485 markedly increased the LC-3II/LC-3I ratio (Figure 3J–L). Autophagy is invariably associated with the conversion of the microtubule-associated protein LC3 from its cytosolic form (LC-3I) to its autophagosome-associated form (LC-3II) [25]. The upregulation of LC3II can be caused by either the up-regulation of autophagosome formation or blockage of autophagic degradation, such as blockage of autophagosome–lysosome fusion. Since MHY1485 has been reported to suppress autophagosome–lysosome fusion [26], these two possibilities cannot be distinguished from each other when cells are treated with MHY1485 alone. To confirm that the increased LC-3II/LC-3I ratio was the result of the accumulation of autophagosomes and not elevated autophagic activity, we measured autophagic flux with the lysosome inhibitor bafilomycin A1, which blocks the fusion of autophagosomes with lysosomes and, therefore, allows us to clamp autophagosome consumption, as previously described [25]. Bafilomycin A1 was added 12hrs before harvesting cells to inhibit lysosomal activity, leading to accumulation of LC3II protein. Western blot analysis showed that LC3II protein increased after treatment of bafilomycin A1, implying the basal autophagic activity. As expected, LC3II protein did not accumulate after treatment with the lysosomotropic reagents, bafilomycin A1, in the cells treated with MHY1485, indicating that MHY1485 did not increase the autophagic flux and suppressed the basal level of autophagic flux (Figure 3J–L). Therefore, MHY1485 indeed inhibited the autophagic process.

Taken together, these results indicate that mTOR inhibition leads to increased EC stemness marked by CD44^+^/CD24^−^ and inducing the mTOR pathway leads to decreased EC stemness. In both cases, autophagy seems to be involved, since an increased autophagy was observed upon mTOR inhibition and there was a reduced autophagy with mTORC1 stimulation.

### 2.3. The mTOR Pathway Down-Regulates Cancer Stemness through Autophagy

Next, we determined whether autophagy plays a key role in mTOR-mediated cancer stemness, using the autophagy inhibitor hydroxychloroquine (HCQ). Concentrations that reduced cell viability by less than 50% (Appendix A) were used for follow-up experiments. Figure 4A–C shows that treatment of esophageal cancer cells with HCQ results in the accumulation of autophagy marker LC-3II, indicating that autophagic flux is blocked. This result suggests that autophagy was successfully inhibited by HCQ. As expected, HCQ leads to a reduced level of the CD44^+^/CD24^−^ CSC population (Figure 4D–F) and the number of spheres formed in 3D culture (Figure 4G–I), suggesting HCQ can inhibit cancer stemness. Of note, Torin-1-mediated cancer stemness up-regulation was significantly reduced when cells were treated with HCQ, suggesting that autophagy plays a crucial role in mTOR-mediated cancer stemness repression.

### 2.4. A CD44^+^/CD24^−^ Subpopulation with CSC Characteristics Is Present in EC Patients-Derived Organoids (ec-PDOs)

To test the clinical relevance of CSC characteristics of CD44^+^/CD24^−^ cells and, subsequently, the treatment of Torin-1 and MHY1485, we generated ec-PDOs from pretreatment EAC biopsies through modifying a culture protocol, based on previous reported studies on Barrett’s epithelium and normal esophageal and EAC cells (Figure 5A,B) [27,28,29]. Enzymatically and mechanically dispersed patient-derived esophageal cancer biopsies were serially cultured as organoids for multiple passages, indicating self-renewal and expansion potential of cancer stem cells (Figure 5A,B). Furthermore, the ec-PDOs stained positive for the EAC marker MUC5AC [30,31], suggesting an EAC origin (Appendix A).

**Figure 4 cancers-14-01806-f004:**
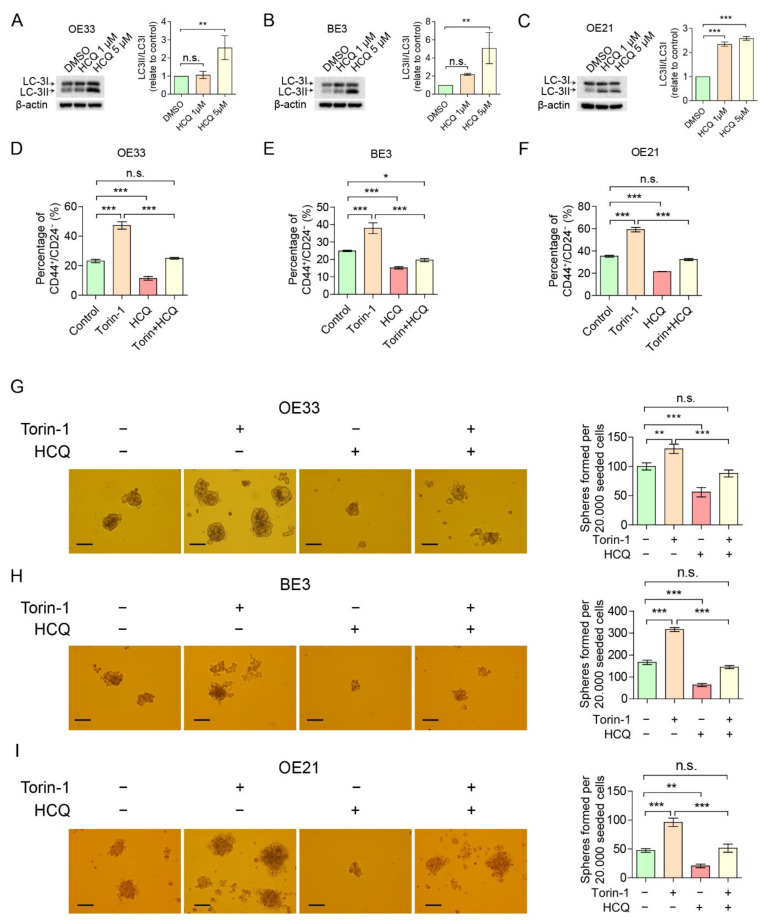
mTOR inhibited cancer stemness via autophgy. (**A**–**C**) Cells treated with HCQ (OE33 at 5 µM, BE3 and OE21 at 1 µM and 5 µM) for 48 h had an increased ratio of LC-3II/LC-3I compared to controls. (**D**–**F**) Flow cytometry analysis of CD44^+^/CD24^−^ in cells treated with a combination of HCQ and Torin-1 or alone (OE33, HCQ at 5 µM and Torin-1 at 4 nM; BE3 and OE21, HCQ at 1 µM and Torin-1 at 8 nM). (**G**–**I**) Sphere formations of cells treated with a combination of HCQ and Torin-1 or alone (OE33, HCQ at 5 µM and Torin-1 at 4 nM; BE3 and OE21, HCQ at 1 µM and Torin-1 at 8 nM). Scale bar = 100 µm. Error bars indicate SD. n.s., not statistically significant; * *p* < 0.05, ** *p* < 0.01 and *** *p* < 0.001 by two-way ANOVA followed by a Tukey–Kramer post hoc test. n.s.: not statistically significant. The uncropped blots are shown in Appendix A.

As soon as the culture generated sufficient tumor organoids, these structures were dissociated into single cells, and FACS analyses were performed to detect the CD44^+^/CD24^−^ CSC population. The percentage of CD44^+^/CD24^−^ cells differed between the patients and showed differences between passages (Figure 5C), indicating the presence of a potential esophageal cancer stem cell population, as seen in our EC cell lines. Since organoid-forming capacity is one of the hallmarks of cancer stemness [32], we assessed this trait by sorting for CD44^+^/CD24^−^ population. After sorting, the CD44^+^/CD24^−^ population showed a higher organoid-forming potential than CD44^+^/CD24^+^ and sorted live population (Figure 6A,B). Another important CSC characteristic is the ability to form secondary organoids after radiation. To this end, we performed radiation experiments on our CD44^+^/CD24^−^ population, CD44^+^/CD24^+^ and live population. The FACS-sorted CD44^+^/CD24^−^ population showed a higher organoid-forming potential after 2Gy compared to the CD44^+^/CD24^+^ population and sorted live population (Figure 6B), indicating a higher radiation resistance. These results further validate our finding that the CD44^+^/CD24^−^ population represents a putative CSC population, which needs to be eradicated.

### 2.5. CD44^+^/CD24^−^ CSC Population in ec-PDOs Can Be Modulated by mTOR

To investigate whether mTOR modulation could affect the percentage of CD44^+^/CD24^−^-expressing cells, we treated sorted CD44^+^/CD24^−^ and CD44^+^/CD24^+^ cells with Torin-1 and MHY1485. After 14 days, ec-PDOs were dissociated into single cells and the resulting percentage of CD44^+^/CD24^−^ cells was assessed. Remarkably, both sorted populations contained less CD44^+^/CD24^−^ CSC population after MHY1485 treatment (Figure 6C and Appendix A). In contrast, Torin-1 enhanced the percentage of CD44^+^/CD24^−^ cells in both sorted populations (Figure 6C and Appendix A). These results indicate that a putative EC stem cell population can be reduced upon mTOR stimulation (Figure 6D).

## 3. Discussion

Hypoxia and nutrient deprivation often occur in advanced cancer, as a result of tumor outgrowing and the available blood supply, thereby contributing to a more aggressive phenotype [33,34,35]. In this study we show that these factors, although probably not even modulated to a reduced level similar to the in vivo situation, may contribute to a more aggressive cancer phenotype by acquiring a more CD44^+^/CD24^−^ CSC-like appearance. This appears to be at least partly regulated through the mTOR pathway and, thereby, potentially activating one of the downstream cellular autophagy mechanisms. Surprisingly, inhibiting the mTOR pathway using Torin-1 led to an increase in the CD44^+^/CD24^−^ CSC population and activating mTOR using MHY1485 resulted in the opposite effect. Moreover, Torin-1-mediated CSCs’ up-regulation is significantly reduced when cells are treated with hydroxychloroquine (HCQ), emphasizing a role for autophagy. Importantly, we also show, in ec-PDOs, that CD44^+^/CD24^−^ cells are more radiation resistant and respond to mTOR inhibition or stimulation in a similar fashion to the cell lines. Thus, mTOR can inhibit cancer stemness through autophagy in esophageal cancer (Figure 6D). Our results are in line with observations in CD133-expressing cancer cells [36], liver cancer cells [37] and hepatocellular carcinoma cells [38], showing that inhibition of the mTOR pathway resulted in increased cancer cell stemness. Matsumoto et al. [36] showed that inhibition of the mTOR pathway up-regulated CD133^+^, another selection marker for CSCs, in 26 cancer cell lines, including gastric, colorectal and lung cancer, but not in EC. Yang et al. [36] showed the same effect in liver cancer cells, also using CD133^+^ as a CSC marker. Hwajung et al. [33] showed that the percentage of CD133^+^EpCAM^+^ cancer stem cell (CSC) populations was significantly increased in hepatocellular carcinoma cell lines Huh7 and HEP3B, when treated with mTOR inhibitors, sirolimus and everolimus. Interestingly, we recently showed an important role for Hedgehog (HH) in EC-CSC regulation [39]. Related to the mTOR signaling pathway, Wang et al. [40] discovered a crosstalk in EC between the mTOR pathway and the HH pathway, also associated with CSCs [40], where an activated mTOR pathway leads to more non-canonical HH pathway activation, through S6K1-mediated Gli1 phosphorylation, thereby releasing Gli1 from its endogenous inhibitor SuFu. Furthermore, elimination of S6K1 activation by mTOR inhibition enhanced the anti-cancer effect of canonical HH pathway inhibition. The authors advocate a combination treatment with HH and mTOR inhibitors to treat EC. It is plausible, however, that when an mTOR inhibitor is used as a single agent, it is targeting the majority population of EC cells, while enriching CSCs. Therefore, despite the potentially smaller proportion of remaining cells after mTOR inhibition, this proportion may demonstrate a more aggressive phenotype.

Our preliminary patients’ data seem to be in line with our cell line results, where indeed, mTOR activation leads to fewer cells with CD44^+^/CD24^−^ expression pattern. Interestingly, a previous paper reported that the inhibition of the mTOR signaling pathway induces the pausing of mouse blastocyst development [41]. These paused blastocysts remain pluripotent and could develop into embryonic stem cells and fertile mice after withdrawal of mTOR inhibition [41]. These results suggest that mTOR is a regulator of differentiation. According to our results, inducing the mTOR pathway down-regulates the percentage of the CD44^+^/CD24^−^ CSC population, also suggesting that the mTOR pathway is involved in EC differentiation. Moreover, activation of oncogenic KRAS is the most common driving event in the self-renewal and maintenance of CSCs [42,43]. However, mTOR, as the activated downstream of RAS signaling, was shown to exert different effects on CSCs in EC cells, in which mTOR can inhibit CSCs in EC cells but KRAS increases CSCs [44,45,46]. These findings indicate that more research is needed to further address the exact role of the mTOR pathway with regards to cancer cells with stem cell properties, to clarify the discrepancy.

Autophagy is a complicated process that consists of autophagosome biogenesis, maturation, and fusion with lysosomes. Several signaling complexes have effects on the respective molecular steps in the regulation of the autophagic process [47]. Autophagy plays an important function in regulating cell homeostasis. Dysfunction of the autophagy machinery often results in a variety of human diseases, such as cancer [47]. Different research groups have extensively studied the well-established regulatory role of mTOR in autophagy [48,49]. We found mTOR can inhibit autophagy, thereby increasing EC-CSCs. This finding is in line with the fact that autophagy is capable of inhibiting CSCs in breast cancer and gastric cancer [50,51], implicating a role for mTOR in the regulation of cancer stem cell properties. In addition, evidence is accumulating that phospholipase D (PLD), an enzyme that promotes the hydrolysis of phosphatidylcholine to generate PA, is a critical regulator of AMPK-mTOR signaling, by a variety of stimuli [52,53]. PLD serves as a new regulator of autophagy, as it can suppress AMPK via mTOR, and AMPK also suppresses PLD activity, implicating PLD as an integral part of energy input to mTOR [54].

Patient-derived organoids (PDOs) have been shown to resemble the tissue/tumor of origin [55] and have been used to explore drug responses [27,56]. Strikingly, we are one of the first groups to explore the field of PDOs in EC, to study potential anti-cancer therapy. Whereas the cell lines OE33 and OE21 contained only CD44^+^/CD24^−^ and CD44^+^/CD24^+^ subpopulations, and BE3 only CD44^+^/CD24^−^ and CD44^−^/CD24^−^ subpopulations, our ec-PDOs exhibited a more heterogeneous population of cells, as demonstrated by our CD44 and CD24 expression FACS plots, suggesting this technique may be superior to cell lines in representing the actual heterogeneity of patients’ tumors.

Torin-1, as the mTOR inhibitor, has been widely used to explore the function of mTOR [57,58]. Our study confirms that inhibiting the mTOR pathway using Torin-1 led to an increase in the CD44^+^/CD24^−^ putative CSC population. In addition, rapamycin also serves as a classical mTOR inhibitor. Since we know rapamycin and Torin act differently [59], and also that rapamycin acts differentially based on the dosage used [60], it is a very meaningful question whether rapamycin would play the same role as Torin-1 here. Moreover, even the concentration of rapamycin could play a role here, but this needs to be explored in the future.

MHY1485 has widely been shown to suppress autophagy signaling by activating mTOR [26,61,62,63]. It has been shown to inhibit autophagy by the inhibition of fusion between autophagosomes and lysosomes, leading to the accumulation of LC3II protein and enlarged autophagosomes [26]. Meanwhile, several studies have suggested that the activation of mTOR signaling provides anti-cancer activity. Llanos et al. [64] reported that hyper-activation of mTORC1 stabilizes p21, and that protein expression of p21 and *p*-S6 (a surrogate for mTORC1 activity) correlates with improved survival in patients with head and neck cancers. Moreover, Han et al. showed that the activation of mTOR signaling using MHY1485 treatment increased the 5-fluorouracil sensitivity of colon cancer cells deficient for p53 [65]. Furthermore, Lue et al. reported that MHY1485 treatment inhibited growth and colony formation, in both cell lines under irradiation and non-irradiation conditions [66]. However, our study is the first that used MHY1485 for anti-tumor stem cells purposes and confirmed that mTOR can suppress CSCs, by inhibiting the autophagic process. MHY1485 might be of therapeutic value in reducing CSCs in the treatment of EC. Moreover, ec-PDOs can potentially represent new possibilities for future anti-cancer research, such as prediction of response to conventional chemoradiation prior to the start of this intense regimen and selection for personalized treatment. In cases where conventional therapy falls short, the use of PDOs could be used to screen for alternative treatment strategies, such as mTOR modulation. In conclusion, mTOR pathway activation leads to less cancer stemness and may serve as a therapeutic target in reducing cancer stemness. Combining mTOR inducer, MHY1485, with conventional chemoradiation could potentially result in a more effective treatment, in which CSCs and the bulk tumor cells are both targeted, leading to both an increased therapy response and a better long-term survival.

## 4. Material and Methods

### 4.1. Cell Lines and Cell Culture

Three EC cell lines were used, BE3 and OE33 both derived from esophageal adenocarcinoma and OE21 derived from esophageal squamous cell carcinoma. BE3 was cultured in GIBCO DMEM medium (Life Technologies, Foster City, CA, USA) while OE33 and OE21 were cultured in GIBCO RPMI 1640 medium (Life Technologies). Media of all cell lines were supplemented with 10% FCS and 1% of penicillin/streptomycin cultured in an incubator of 5% CO_2_ and 37 °C. All cell lines were passaged at 70% confluency.

### 4.2. Low-Oxygen and Nutrient Deprivation Experiments

Low-oxygen experiments were performed in a hypoxia chamber containing 5% O_2_, while control cells which were cultured under normal culture conditions with 21% O_2_, for a duration of 48 h followed by harvesting the cells for flow cytometric analysis. Cells were starved with HBSS solution containing 1% rich medium (DMEM + 10% FBS) overnight [67]. Downstream targets of the mTOR pathway were checked using antibodies directed against mTOR (1:1000, Sigma-Aldrich, Louis, MO, USA) and *p*-mTOR (1:1000, Sigma-Aldrich). SOX2 antibody was purchased from Cell Signaling (1:1000). Each western blot experiment was performed at least three times. The most representative blot is shown.

### 4.3. Fluorescence-Activated Cell Sorting (FACS)

Flow cytometric analysis of the expression of CD44 and CD24 was performed on FACS-Calibur (BD Biosciences, Franklin Lakes, NJ, USA). CD24 antibody conjugated with FITC (BD Biosciences) and CD44 antibody conjugated with PE (BD Biosciences) along with their corresponding isotype controls were used. CD44^+^/CD24^−^ cells and CD44^+^/CD24^+^ cells were sorted in OE33 and OE21 cells using the same antibodies with MoFlo Asterios or MoFlo-XDP cell sorter (Beckman Coulter). The most extreme (3–15%) population of each subpopulation was sorted based on our previous studies [5,39].

### 4.4. Cell Viability Assay

The cell viability was examined by MTT assay. Cells were seeded in 96-well plates at a density of 2000 cells/100 uL and treated with different concentrations of Torin-1 (4, 8, 16, or 32 nM), MHY1485 (0.5, 1, 2, or 4 μM) or HCQ (1, 5, 10, or 20 μM) for 48 h. Then MTT (Sigma-Aldrich) solution was added to each well at the endpoint. After incubation for 3 h, the optical density (OD) values were measured with a microplate reader set at 490 nm.

### 4.5. mTOR Pathway Modulation

mTOR inhibitor Torin-1 (Tocris Bioscience, Bristol, UK), and mTOR inducer MHY1485 (Sigma-Aldrich) were used to modulate the mTOR pathway. Cells were treated 48 h with either an inhibitor or an activator of the mTOR pathway. The autophagy inhibitor hydroxychloroquine (HCQ) (Sigma-Aldrich) was used to inhibit autophagy. Cells were treated with HCQ for 48 h to block autophagy as previously described [68]. After treatment, cells along with their corresponding control cells (DMSO) were harvested for western blots, flow cytometry and/or sphere forming assays. *p*-mTOR (1:1000, Cell Signaling, Danvers, MA, USA), mTOR (1:1000, Cell Signaling) and LC3II (1:1000, Novus Biologicals, LLC, Englewood, CO, USA) markers for autophagy, which are downstream of the mTOR pathway, were measured by western blot. Each western blot experiment was performed a minimum of three times. The most representative blot is shown. Cells of either treated or untreated (DMSO) population were seeded in triplicate in 2 mL Mammocult medium with supplement (Stem cell technologies) of a 6-well plate coated with agarose to prevent the attachment of the cells onto the bottom of the plate. The concentration of agarose was 0.023 g/mL dissolved in distilled water. Following this, 0.2 mL Mammocult was added each day to nourish the cells. Images of spheres were made under the microscope after 5 days and the number of spheres was subsequently counted. Patient material was sorted into CD44^+^/CD24^−^ CSC-like population vs. CD44^+^/CD24^+^ non-CSC populations. After sorting, the lowest obtained number of cells of the two sorted populations were seeded for each population in Matrigel in a 12-well plate to generate organoids with medium containing either 10 nM Torin-1, 1 µM MHY1485 or DMSO (controls). Medium in the presence of the drugs or DMSO was changed every third day. After 14 days (1 passage), the number of organoids was counted and the generated organoids were digested into single cells and underwent FACS analyses to check for the expression of CD44 and CD24.

### 4.6. Patient Material and Patient-Derived Organoid Culture

This study has been approved by our Medical Ethical Committee. Patients without nCRT and proven EC that visit our center for an endoscopic ultrasonography of the esophagus for disease staging during 2015–2021 with a signed informed consent were included. In case of a visible tumor, six biopsies at different sites of the tumor were taken by an experienced gastroenterologist and collected in a 50 mL tube containing 5 mL HBSS (GIBCO) on ice. Digestion of the tissue was performed after cutting the tissue manually with a pair of scissors in a gentleMACS C tube (Miltenyi Biotec, Bergisch Gladbach, Germany) containing 62.5 µL collagenase I (3.5 mg/100 µL, GIBCO), 62.5 µL dispase (8 mg/100 µL, GIBCO) and 625 µL CaCl_2_ (Sigma-Aldrich) in 2.5 mL HBSS 1% BSA followed by an incubation in a moving water bath at 37 °C for 5–30 min. Afterwards, the tissue was cut again manually with scissors followed by a second digestion step in a moving water bath at 37 °C for another 5–30 min. The digestion was then terminated by adding 3 mL HBSS 1% BSA. After spinning down (400 g, 5 min), the pellet was resuspended in 1 mL HBSS 1% BSA and filtered through a yellow filter (100 µm) into a 50 mL tube. The remaining tissue on the filter was pushed through as much as possible by the cap of a 5 mL syringe. The C-tube and the cap of the syringe was then washed with another 1 mL HBSS 1% BSA to ensure no waste of tissue. This suspension was then centrifuged followed by resuspending the pellet in esophageal (ES) medium and then mixed with Matrigel (Corning, Two Oak Park, MA, USA) in a 1:2 fashion (25 µL ES medium + 50 µL matrigel) per well in a 12-well plate to generate organoids. Depending on the size of the pellet 1–3 wells were seeded. After 20 min, allowing the Matrigel to solidify, 1 mL of ES medium was added to each well. After 3 days of the initial passage (p0), cells were passaged into the first passage (p1) followed by the next passage after another 11 days (p2). Afterwards, each passage will take place after 14 days. Matrigel was disrupted by adding 1 mL DMEM F12 (GIBCO) containing dispase (1 mg/mL, GIBCO) to each well with an incubation duration of 45–60 min at 37 °C. Next, 100 µL aliquot was taken to count the number of organoids. After collection in a tube, the wells were washed once with 2 mL PBS/0.2% BSA and collected in the tube. The organoids were then spun down (400 g, 5 min) and resuspended in 1 mL of trypsin, spread back over the original culture wells and incubated at 37 °C. Every 5 min the organoid dissociation was checked under the microscope. Trypsinization was ended by adding the dissociated organoids into a tube containing 1 mL of PBS/1% BSA. The wells were washed with 1 mL of PBS/1% BSA and also collected into the tube followed by spinning down (400 g, 5 min). Pellets were resuspended in 1 mL of ES medium and cells were counted with trypan blue. Further, 10,000 cells per well were seeded into the next passage. The ES medium contains 40% DMEM F12 with 1% penicillin/streptomycin and 1% glutamax, 50% WNT medium and 10% R-spondin medium supplemented with the following growth factors and reagents: 1 × HEPES (10 mM, Life Technologies), nicotinamide (10 mM, Sigma-Aldrich), Noggin (25 ng/mL, Preprotech), B27 (10 uL/mL, GIBCO), N2 (10 µL/mL, GIBCO), EGF (50 µg/mL, Sigma-Aldrich), FGF10 (10 µL/mL, Preprotech), A8301 (1 µL/mL, Tocris Biosciences), *n*-acetyl cysteine (1 mM, Sigma-Aldrich), Gastrin (10 nM, Tocris Bioscience), Primocin (1 mg/mL, InvivoGen), penicillin/streptomycin (GIBCO) 1% and Y-27632 (10 uL/mL, all passages except p0, Abcam). WNT and R-spondin producing cell lines to produce WNT and R-spondin conditioned media were kindly given by Prof. Dr. H. Clevers.

### 4.7. Immunohistochemistry Staining

The ec-PDOs were fixed with 4% formaldehyde and embedded into paraffin. Five micrometer paraffin sections were dewaxed and boiled for 8 min with pre-heated antigen retrieval buffer. Subsequently, sections were stained with antibody against MUC5AC (Cat. No. SC-21701 Santa Cruz). Visualization for bright field microscopy was accomplished by adding specific secondary biotin carrying antibody at 1:300 dilution. Nuclear counterstaining was performed with hematoxylin. Sections incubated with immunoglobulin (Ig) G of appropriate species (mouse) as the primary antibody were used as negative controls.

### 4.8. Radiation Experiments

After sorting cells of ec-PDOs into CD44^+^/CD24^−^ CSC population and CD44^+^/CD24^+^ non-CSC population, equal amounts of cells were seeded in ES medium mixed with Matrigel. After the Matrigel solidified and 1 mL of ES medium was added, plates were irradiated with 2 Gy (Cesium source). The amounts of organoids were counted after 14 days and compared to the non-irradiated control.

### 4.9. Statistical Analysis

The comparisons between two groups were performed with Student’s *t*-tests and those among >2 groups with one-way ANOVA followed by appropriate multiple comparison tests using SPSS statistics 20.0 software. A *p*-value < 0.05 was considered as significant, and all tests were 2-sided.

## 5. Conclusions

This study reveals that mTOR can suppress CSCs via autophagy in esophageal cancer. This finding has important clinical implications and can trigger new mechanistic studies on the role of mTOR toward developing new therapeutic strategies against esophageal CSCs.

## Figures and Tables

**Figure 1 cancers-14-01806-f001:**
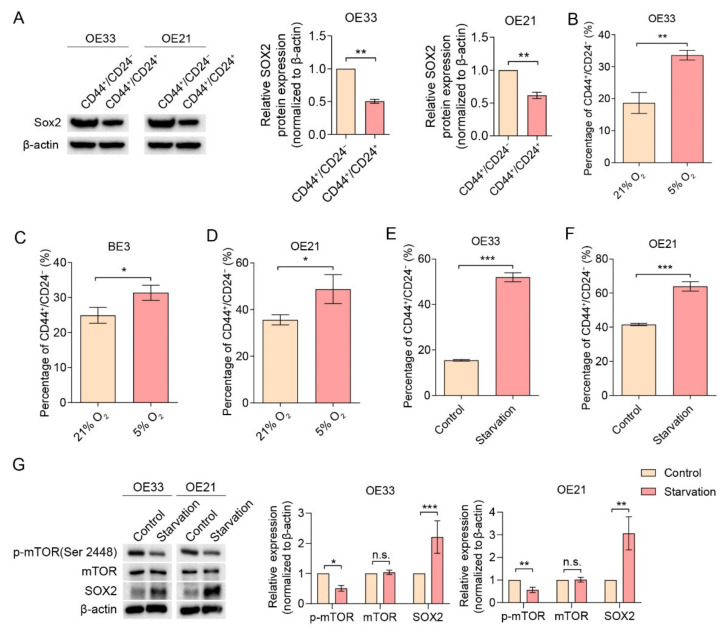
Reduced oxygen and nutrient deprivation selects for CD44^+^/CD24^−^ CSC population and coincides with reduced mTOR phosphorylation. (**A**) CD44^+^/CD24^−^ CSC population expresses increased SOX2 compared to CD44^+^/CD24^+^ population. (**B**–**D**) CD44^+^/CD24^−^ CSC population is favored in esophageal cancer cell lines under low-oxygen (5% O_2,_ 48 h) environment compared to cells cultured under normoxic conditions (*n* = 3). (**E**,**F**) CD44^+^/CD24^−^ CSC population is enriched under nutrient-deprived conditions (48 h) compared to normal culture medium under normoxic conditions (*n* = 3). (**G**) Starved cells exhibit a decreased expression of *p*-mTOR and greater expression of SOX2. Error bars indicate SD. * *p* < 0.05, ** *p* < 0.01 and *** *p* < 0.001 by student’s *t*-test. n.s.: not statistically significant. The uncropped blots are shown in Appendix A.

**Figure 2 cancers-14-01806-f002:**
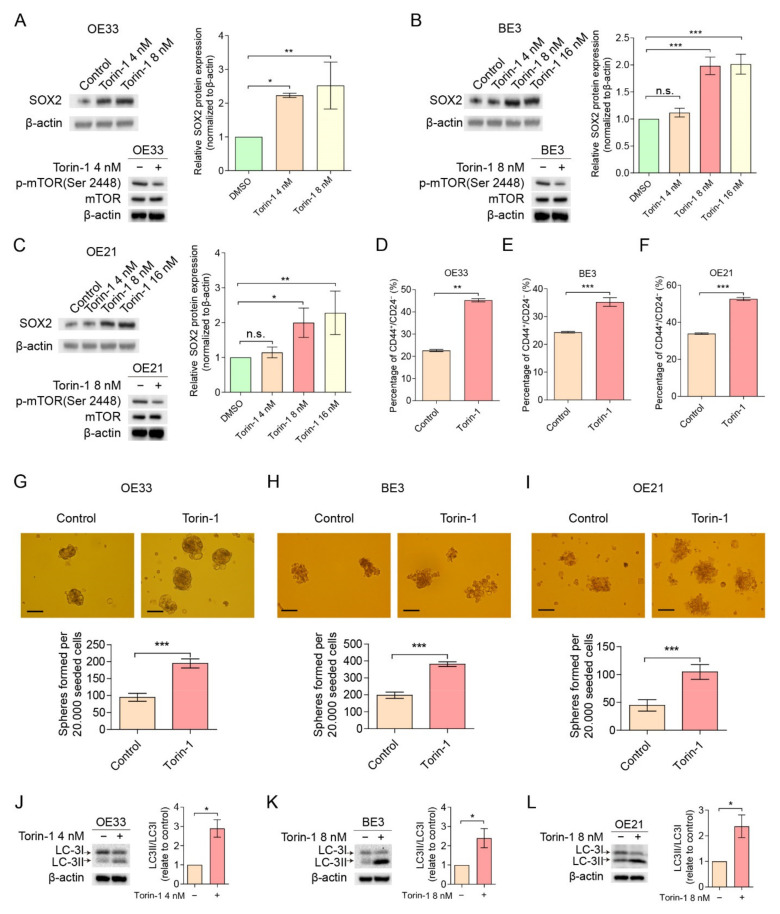
mTOR inhibitor Torin-1 up-regulates CD44^+^/CD24^−^ CSC population, sphere formation and autophagy. (**A**–**C**) Cells treated with Torin-1 showed an increased expression of SOX2 compared to DMSO controls (OE33 at 4 nM, BE3 and OE21 at 8 nM and 16 nM). (**D**–**F**) Cells treated with Torin-1 (OE33 at 4 nM, BE3 and OE21 at 8 nM) for 48 h showed an up-regulation in the CD44^+^/CD24^−^ CSC population compared to DMSO controls (*n* = 3). (**G**–**I**) Cells treated with Torin-1 (OE33 at 4 nM, BE3 and OE21 at 8 nM) for 48 h elevated the sphere generation capacity compared to DMSO controls (*n* = 3). Scale bar = 100 µm. (**J**–**L**) Cells treated with Torin-1 (OE33 at 4 nM, BE3 and OE21 at 8 nM) for 48 h increased the ratio of LC-3II/LC-3I compared to DMSO controls. Error bars indicate SD. n.s., not statistically significant; * *p* < 0.05, ** *p* < 0.01 and *** *p* < 0.001 by two-way ANOVA followed by a Tukey–Kramer post hoc test or student’s *t*-test, where appropriate. n.s.: not statistically significant. The uncropped blots are shown in Appendix A.

**Figure 3 cancers-14-01806-f003:**
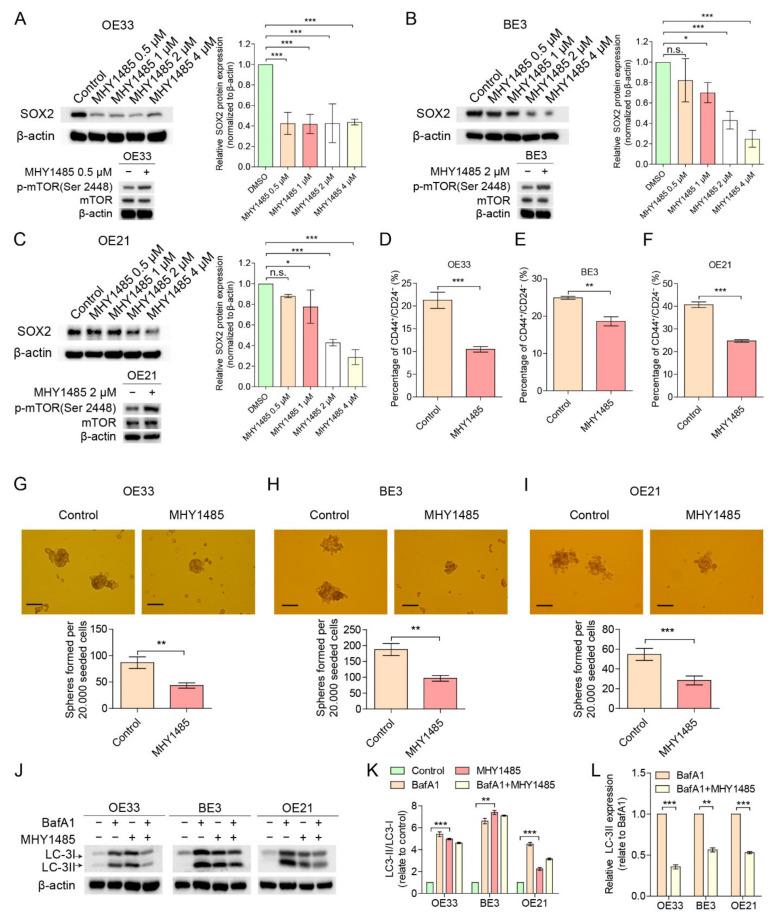
mTOR inducer MHY1485 down-regulates CD44^+^/CD24^−^ CSC population, sphere formation and autophagy. (**A**–**C**) Cells treated with MHY1485 showed a decreased expression of SOX2 compared to DMSO controls (OE33 ≥ 0.5 µM, BE3 ≥ 1 µM and OE21 ≥ 2 µM). (**D**–**F**) Cells treated with MHY1485 (OE33 at 0.5 µM, BE3 at 1 µM and OE21 at 2 µM) for 48 h showed a down-regulation of the CD44^+^/CD24^−^ CSC population compared to DMSO controls (*n* = 3). (**G**–**I**) Cells treated with MHY1485 (OE33 at 0.5 µM, BE3 at 1 µM and OE21 at 2 µM) for 48 h had a decreased sphere generation capacity compared to DMSO controls (*n* = 3). Scale bar = 100 µm. (**J**–**L**) Western blots analysis of LC-3II/LC-3I in cells treated with MHY1485 (OE33 at 0.5 µM, BE3 at 1 µM and OE21 at 2 µM) or control for 48 h followed by either 10 μM bafilomycin A1 or DMSO for an additional 4 h. Error bars indicate SD. n.s., not statistically significant; * *p* < 0.05, ** *p* < 0.01 and *** *p* < 0.001 by two-way ANOVA followed by a Tukey–Kramer post hoc test or student’s *t*-test, where appropriate. n.s.: not statistically significant. The uncropped blots are shown in Appendix A.

**Figure 5 cancers-14-01806-f005:**
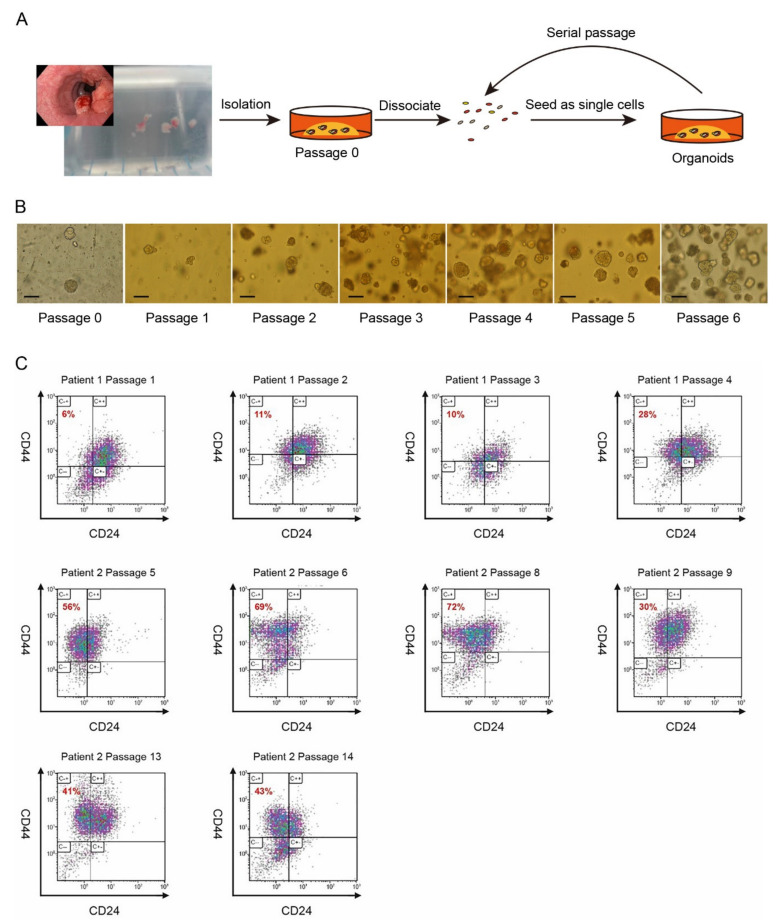
CD44^+^/CD24^−^ is present in EC patient-derived Organoids (ec-PDOs). (**A**) Scheme showing an overview of the isolation and expansion of ec-PDOs. (**B**) Representative images of ec-PDOs in different passages. Scale bar = 100 µm. (**C**) Representative FACS plots of CD44^+^/CD24^−^ expression in different ec-PDOs in different passages.

**Figure 6 cancers-14-01806-f006:**
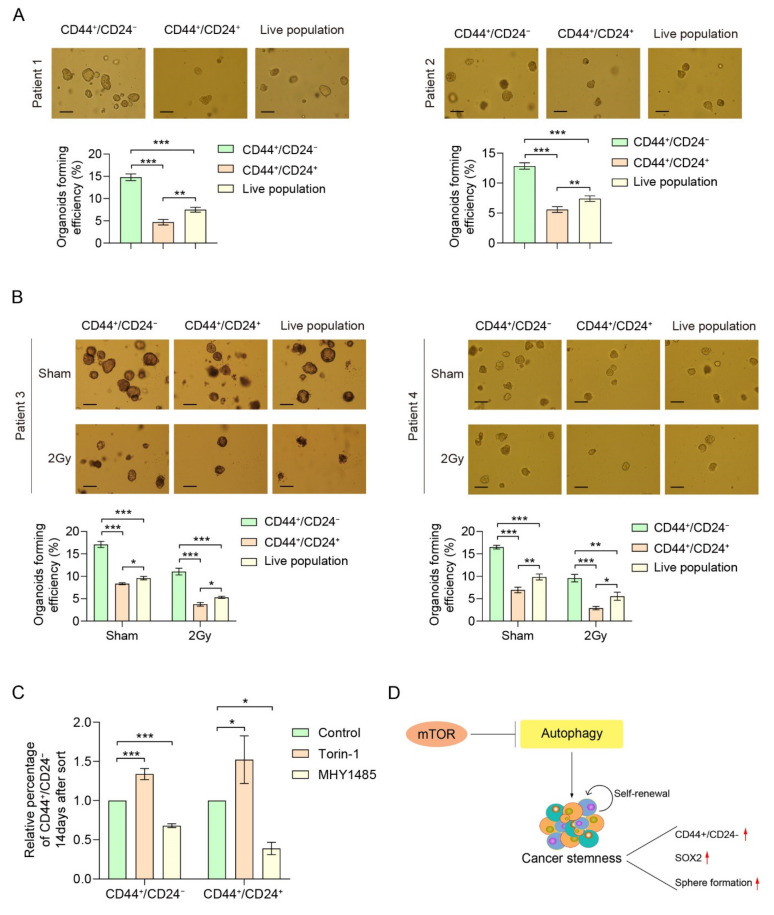
CD44^+^/CD24^−^ represents a CSC population and can be modulated by the mTOR pathway. (**A**,**B**) CD44^+^/CD24^−^ population forms more organoids compared to CD44^+^/CD24^+^ or CD44^−^/CD24^−^ populations and are more radiation-resistant upon 2 Gy radiation. Scale bar = 100 µm. (**C**) Both CD44^+^/CD24^−^ and CD44^+^/CD24^+^ -sorted cells treated with Torin-1 (10 nM, 14 days) showed an increased CD44^+^/CD24^−^ phenotype compared to DMSO controls whereas the same sorted populations show a more pronounced decrease in CD44^+^/CD24^−^ cells with MHY1485 (1 µM, 14 days). (**D**) Schematic representation of the role of mTOR in esophageal cancer stemness through autophagy. Error bars indicate SD. * *p* < 0.05, ** *p* < 0.01 and *** *p* < 0.001 by two-way ANOVA followed by a Tukey–Kramer post hoc test.

## Data Availability

Not applicable.

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
