# Peer review of "Role of mTOR through Autophagy in Esophageal Cancer Stemness"

_cancers, 2022, doi:10.3390/cancers14071806_

Round 1

Reviewer 1 Report

Article by Dr. Coppes and group elaborating the novel role of mTOR through autophagy in cancer stemness. It is very well designed experimental manuscript which follows the hypothesis of the scientific question asked in this manuscript. Though few things need to be addressed before it is ready for acceptance. They are as follows:

  1. Authors should add a model depicting the thesis of this work, elaborating the theme of this manuscript. This can be added as an sub-figure and should be discussed in the discussion section.
  2. The authors should mention the role of mTOR and AMPK in autophagy in introduction part and also there is a missing link- PLD/Phospholipase D which interconnects mTOR and AMPK for their significant role in autophagy ( PMID: 25632961 and PMID: 24317201). This aspect should be discussed which will give a more detailed view of mTOR's role in autophagy in cancer development.  
  3. It is provocative to enquire authors perspective of using rapamycin-the classical mTOR inhibitor in place of Torin-1. Since we know rapamycin and Torin acts differently (PMID: 29063293) and also Rapamycin acts differentially based on the dosage used (PMID: 26916116 ). Authors should mention if they have tried of using rapamycin or any rapalogs instead of Torin-1 in this study. Otherwise few lines should be added discussing authors point of view if rapamycin would play same role what Torin-1 did here, also in what concentration of rapamycin would do that. This addition will be efficient to get a holistic view of mTOR inhibition in autophagy-specifically in cancer stemness.
  4. Authors should add few lines discussing the role of mTOR as downstream of KRAS which in turn also affecting the stemness of cancer (PMID: 29311260 and PMID: 24231729). This addition will be relevant of the scope of this study in broader aspect. 

Reviewer 2 Report

The manuscript by Du and colleagues analysed the role of autophagy in stemness maintenance in oesophageal cancer stem cells. This is something. That has been already described for multiple kind of cancers and the results support the role of autophagy in the maintenance of CSCs also in this case.

I’ve just some comments about the presentation of the results and a couple of doubts the authors should address.

Results section:

page2, line 115: the authors should state which kind of cells are BE3, and which differs from OE33 and OE21.

In addition, the supplementary figure, which should address the CD44+/CD24- status of the cell lines is not clear at all, as there is no reference, i.e., unstained double negative populations, and they seem to express both the markers at high levels. So how the “right and left populations” have been chosen?

Line 117: what the Supplementary figure 1C-E should address? There are just cells under “normoxic” conditions and no data of the reduced oxygen conditions.

Line 119: what does it mean “hypoxia poor vascularization”?

Lines 123-124: It is not clear whether this refers to starvation alone or starvation in reduced oxygen conditions?

Figure 2: WB images should be “harmonized”, which means the same doses for all the different cell lines, and the p-mTOR for all the doses. In addition, better to have the WB on one side and the relative quantifications on the other. As for the percentage of CD+/- the same comment as above states, flow cytometry panels would be welcome. Panels J-L the same comment as above for panels A-F.

Figure 3. The same comments about the dispositions of panels and graphs as in figure 2.

Panels K-L I’m not convinced at all by the data referring to the relative increase of the ratio between LC3II and LC3I, as by the images it looks like that this does not change following treatments. Bafilomycin alone treatment should lead to a marked increase of the ratio, while MHY1485 alone should decrease, if it works as an autophagy inhibitor, in respect to untreated controls. In addition, the timing of the experiment looks like to be too long to observe autophagy induction.

Page 7, line 237: autophagy is not knocked down, rather the autophagic flux is blocked, by chloroquine administration, which by the way is not the best reagent to use for blocking autophagy at time longer than 8-10 hours, because of its side effects.

Figure 5 and relative text. Which are the criteria used for stating that those are PDOs? Which are the markers tested to prove that they are organoids? They rather looks like spheres.

Round 2

Reviewer 1 Report

All concerns have been addressed, ready for acceptance. 

Reviewer 2 Report

--